# Cardiac Relapse of Acute Lymphoblastic Leukemia Following Hematopoietic Stem Cell Transplantation: A Case Report and Review of Literature

**DOI:** 10.3390/cancers13225814

**Published:** 2021-11-19

**Authors:** Irtiza N. Sheikh, Dristhi Ragoonanan, Anna Franklin, Chandra Srinivasan, Bhiong Zhao, Demetrios Petropoulos, Kris M. Mahadeo, Priti Tewari, Sajad J. Khazal

**Affiliations:** 1Division of Pediatrics and Patient Care, University of Texas MD Anderson Cancer Center, Houston, TX 77030, USA; isheikh1@mdanderson.org; 2Department of Pediatrics, Pediatric Stem Cell Transplantation and Cellular Therapy, CARTOX Program, University of Texas at MD Anderson Cancer Center, Houston, TX 77030, USA; DRagoonanan@mdanderson.org (D.R.); dpetro@mdanderson.org (D.P.); KMMahadeo@mdanderson.org (K.M.M.); PTewari@mdanderson.org (P.T.); 3Center for Cancer and Blood Disorders, Children’s Hospital Colorado, Aurora, CO 80045, USA; Anna.Franklin@childrenscolorado.org; 4Cardiac Center, Children’s Hospital of Philadelphia, Philadelphia, PA 19104, USA; srinivasac@chop.edu; 5Department of Pathology and Laboratory Medicine, The University of Texas Health Science Center McGovern Medical School, Houston, TX 77054, USA; Bihong.Zhao@uth.tmc.edu

**Keywords:** relapse/refractory leukemia, extramedullary disease, stem cell transplantation, CAR T-cell therapy, cardiac relapse, pediatrics

## Abstract

**Simple Summary:**

We present a case of a pediatric patient with symptomatic leukemia involving the heart without bone marrow involvement. The case is followed by a literature review of advances in leukemia treatment including immunotherapy, CAR T-cell therapy, improvement in the treatment of graft-versus-host disease, and the applicability of these treatments to leukemia involving areas outside of the bone marrow. We also explore the feasibility of treatments such as CAR T-cell therapy and blinatumomab and the associated risks for treating patients with leukemia infiltrating the heart.

**Abstract:**

Isolated extramedullary relapse of acute lymphoblastic leukemia (ALL) occurs in soft tissues and various organs outside the testis and central nervous system. Treatments such as hematopoietic stem cell transplantation and more novel modalities such as immunotherapy have eradicated ALL at extramedullary sites. In some instances, survival times for relapsed ALL at these sites are longer than those for relapsed disease involving only the bone marrow. Isolated relapse of ALL in the myocardium is rare, especially in children, making diagnosis and treatment of it difficult. More recent treatment options such as chimeric antigen receptor T-cell therapy carry a high risk of cytokine release syndrome and associated risk of worsening cardiac function. Herein we present the case of an 11-year-old boy who presented with relapsed symptomatic B-cell ALL in the myocardium following allogeneic hematopoietic stem cell transplantation. This is an unusual presentation of relapsed ALL and this case demonstrates the associated challenges in its diagnosis and treatment. The case report is followed by a literature review of the advances in treatment of pediatric leukemia and their application to extramedullary relapse of this disease in particular.

## 1. Introduction

Isolated symptomatic extramedullary relapse (EMR) in the myocardium is an uncommon phenomenon that has been described in rare instances in adult and pediatric acute lymphoblastic leukemia (ALL) patients. The literature includes a report of a 47-year-old man with isolated B-cell ALL relapse involving the myocardium who eventually died of cardiac arrest secondary to heart failure and complete atrioventricular (AV) blockage and a 5-year-old girl with initial isolated cardiac involvement of T-cell ALL but eventual progression and death [1,2]. Given the rarity of isolated EMR of ALL in the myocardium, standard management guidelines for patients with it are lacking. Treatments such as chimeric antigen receptor (CAR) T-cell therapy and post-transplant donor lymphocyte infusion (DLI) are useful in managing EMR, but their specific efficacy against relapses involving the heart is unclear, and it may be associated with unfavorable toxicity profiles [3]. Cytokine release syndrome (CRS), localized inflammation, and pseudoprogression may limit the use of immunotherapies in this setting. Difficulty in arriving at an early biopsy-proven diagnosis of EMR of ALL, a small number of patients with the occurrence, a lack of effective therapy with a favorable side effect profile, and prior exposure to high doses of cardiotoxic chemotherapy and radiation make managing patients with isolated ALL relapse involving the heart difficult. Here we present the case of an 11-year-old boy with symptomatic relapse involving the myocardium and highlight the diagnostic and treatment challenges, as well as application of immunotherapies, in relapsed ALL in the myocardium.

## 2. Case Presentation

We present an 11-year-old African American boy who initially presented with recurrent right thigh and cheek pain and numbness. A bone marrow (BM) aspiration and biopsy confirmed the diagnosis of Philadelphia chromosome-negative B-cell ALL. The clinical characteristics, including initial lab results, are summarized in Table 1. The patient had a right facial droop prior to starting induction chemotherapy, classifying him as CNS3c ALL due to clinical signs of neurological involvement of the leukemia [4]. He was enrolled on the very high-risk arm of the Children’s Oncology Group AAL1131 study [5]. The patient’s treatment regimens are summarized in Table 2. The patient had resolution of the facial droop and improvement in right thigh pain shortly after the initiation of induction chemotherapy. He experienced minimal residual disease (MRD)-negative (by flow cytometry) complete remission at the end of induction chemotherapy with no evidence of leukemic cells in cerebrospinal fluid analysis. Due to technically challenging lumbar punctures and frequent need for intrathecal chemotherapy, the patient underwent ommaya reservoir placement shortly after the end of induction chemotherapy. Prior to this, he underwent magnetic resonance imaging (MRI) of the spine due to back pain, which demonstrated possible diffuse leukemic infiltration of the lumbar spine marrow versus chemotherapy effects and atypical hemangiomas within the L2 and L3 vertebrae as the likely causes of the back pain. However, the scan showed no leptomeningeal enhancement.

The patient proceeded to consolidation chemotherapy. He experienced cranial nerve deficits characterized by right facial numbness, intermittent double vision, unusual sensation and taste, and excessive salivation. He also complained of worsening and persistent right thigh pain. Due to concern for leptomeningeal disease, the patient underwent MRI of the brain and the spine. Imaging of the brain was unremarkable, but that of the spine showed resolution of prior enhancements, a decrease in a prior enhancing focus at the L3 vertebra, and nothing suspicious for leukemia involvement. Cerebrospinal fluid analysis was normal, as well. Due to extensive clinical neurological involvement of his leukemia, the patient received craniospinal irradiation of the brain at 18 Gy in 10 fractions followed by proton therapy to the spine at 15 cobalt gray equivalents in 10 fractions and chemotherapy according to Pediatric Oncology Group protocol POG9412 [6]. After completion of craniospinal irradiation, the patient had very early combined marrow and central nervous system relapse within 6 months of diagnosis and was transitioned to salvage chemotherapy. He underwent a reinduction regimen consisting of hyper-CVAD [7]. Shortly after starting reinduction chemotherapy, the patient had partial neurological function improvement as demonstrated by improved sensation in the distribution of maxillary and mandibular segments of cranial nerve V. Eventually, the patient experienced MRD-negative complete remission following two cycles of hyper-CVAD and had no malignant cells in cerebrospinal fluid analysis.

The patient proceeded to total-body irradiation-based myeloablative hematopoietic stem cell transplantation (HSCT) with a single unrelated umbilical cord blood unit (4/6 human leukocyte antigen match, one A and one B antigen mismatch, sex and ABO match). Graft-versus-host disease (GVHD) prophylaxis consisted of tacrolimus and mycophenolate mofetil. BM evaluation on day +38 after HSCT showed 6% blasts according to morphology, with flow cytometry showing 2.5% leukemic blasts, indicative of residual acute leukemia. A repeat BM aspiration on day +51 following HSCT revealed MRD-negative remission. Treatment with mycophenolate was discontinued on day +57, but that with tacrolimus was continued. Due to decreased appetite and concern for acute GVHD, the patient underwent upper and lower endoscopy on day +122 and was noted to have grade II acute GVHD of the upper and lower gastrointestinal system that was managed with systemic corticosteroids.

About 8 months after HSCT, the patient was admitted to our hospital for fever, shortness of breath, and a chest X-ray showing significant pleural effusions requiring thoracentesis. He was admitted to the pediatric intensive care unit, where he underwent endotracheal intubation due to acute respiratory failure. He experienced rapidly progressive bradycardia with loss of consciousness requiring cardiopulmonary resuscitative efforts. An echocardiogram revealed echogenic areas over the basal interventricular septum indicative of infiltration (Figure 1) with mildly decreased global left ventricular (LV) systolic function (LV shortening fraction of 25% and LV ejection fraction of 48%), mild LV hypertrophy, and a normal LV cavity size. He also had evidence of impaired myocardial systolic contractility and diastolic relaxation indicated by low LV tissue Doppler velocities. The right ventricle exhibited normal size, wall thickness, and systolic function. The patient had changes in AV conduction over a short interval that progressed to complete AV blockage with a variable wide QRS complex escape rhythm at an average rate of 30–40 bpm over 4–5 h (Figure 2). The patient was urgently transferred to a dedicated cardiovascular intensive care unit. After acute resuscitation and stabilization, physicians placed a temporary dual chamber transvenous pacemaker system with an external pulse generator via right internal jugular vein access. At the time of this procedure, physicians obtained right ventricular endomyocardial biopsy samples from the apical septum. The patient promptly recovered following establishment of cardiac pacing with stable and satisfactory hemodynamics.

A BM biopsy showed trilineage hematopoiesis with no evidence of leukemia, cerebrospinal fluid analysis was negative for malignant cells, and peripheral blood analysis showed 100% donor chimerism. However, a right ventricular endomyocardial biopsy showed diffuse lymphoid involvement indicative of ALL infiltration of the right ventricular myocardium (Figure 3). Immunohistochemical stains of the endocardial biopsy sample obtained from the patient demonstrated diffuse and severe lymphoid infiltrates. Staining for CD3, CD10, CD20, CD34, CD79a, PAX-5, CD99, CD1a, and terminal deoxynucleotide (TdT) was performed with adequate controls. The results showed the infiltrating cells were nearly exclusively positive for the B-lymphocyte marker PAX-5, with a few positive for TdT. Polymerase chain reaction analysis of the endomyocardial biopsy for a viral panel was negative, which ruled out viral myocarditis. These results supported the diagnosis lymphoblastic leukemia. The patient underwent a positron emission tomography (PET) scan, the findings of which were compatible with multifocal ^18^F-fluorodeoxyglucose-avid disease involving the sphenoid bone of the skull, right femoral diaphysis and sacrum, soft tissue of the right orbit, right and left ventricles of the heart, mesentery, and retroperitoneum (Figure 4). Computed tomography of the orbits indicated an area of increased attenuation of the superior lateral orbit, causing concern for leukemic infiltration. Following this ALL relapse, the patient began receiving chemotherapy with R-MOAD [8].

The patient underwent placement of a permanent dual-chamber transvenous pacemaker system without complications 3 weeks after completion of intensive chemotherapy due to persistent AV blockage. Subsequent echocardiograms showed improvement in and near-normalization of ventricular systolic function (Figure 5A).

A BM biopsy at the end of one cycle of R-MOAD chemotherapy demonstrated MRD-negative complete remission. However, the patient continued to have residual myocardial disease based on PET/computed tomography (Figure 4). Shortly after completion of the first cycle of R-MOAD chemotherapy, he was readmitted to the pediatric intensive care unit due to concerns about acute cardiac failure (Figure 5B). He had evidence of cardiomyopathy with moderately depressed global ventricular systolic function (LV ejection fraction, 35%). The patient was transitioned to a palliative chemotherapy regimen consisting of intrathecal cytarabine, intravenous etoposide, and intravenous cyclophosphamide. He was eventually discharged home but returned due to refractory cardiogenic shock requiring intensive care management. More than 13 months after transplantation, the patient’s AV conduction did not recover, and he remained chronically paced for more than a month until he died of cardiac arrest secondary to heart failure due to severe LV systolic dysfunction 412 days after HSCT.

## 3. EMR of ALL

No large-scale studies have examined the prevalence of leukemic infiltration of the myocardium in pediatric ALL cases in the antemortem period. In postmortem examination, nearly 40% of pediatric patients with myeloid or lymphoblastic leukemia had focal myocardial infiltration. However, 95–97% of the patients in the study were not noted to have cardiac or respiratory symptoms, including signs of congestive heart failure, when they were alive [9,10]. Unlike extramedullary leukemic involvement of the liver, spleen, central nervous system, and testis in initial and relapsed leukemia cases, symptomatic leukemic infiltration of the myocardium remains rare in the pediatric population. When known cardiac infiltration of leukemia in these patients has occurred, it has been associated with systemic initial presentation or relapse characterized by the presence of peripheral blasts and blasts in the BM leading to a diagnosis of B-cell ALL [11,12]. In one report of cardiac relapse of acute leukemia in a 14-year-old female patient, the initial diagnosis was T-cell ALL, which is more aggressive and has a worse prognosis than other types of ALL [2,13]. The cardiac relapse was followed 5 months later by progression [2]. Diagnosis of an isolated symptomatic cardiac relapse is also difficult due to the location of the ALL infiltration, need for myocardial biopsy, and/or need for pericardial fluid analysis [11,14,15]. Signs and symptoms of cardiac relapse of ALL such as chest pain, difficulty breathing, presence of pleural or pericardial effusion, and hypertrophic cardiomyopathy may be nonspecific and difficult to distinguish from chemotherapy-, irradiation-, or other treatment-related cardiotoxic effects [1,16,17,18]. Initial evaluation of cardiac ALL relapse includes an electrocardiogram, an echocardiogram, and imaging, such as a chest X-ray, PET, and dedicated cardiac computed tomography or MRI [1,2,11,12,17,19]. Tissue samples such as endocardial biopsy samples and pericardial fluid samples collected for cytology should be analyzed using flow cytometry for immunophenotyping [2,14].

Soft tissue appears to be a common site of EMR in patients with relapsed ALL, especially after HSCT [20]. Areas such as the breast, kidneys, intestines, and liver are noted to be the most common sites of EMR after the testis and central nervous system [21,22]. Rare sites of EMR in the pediatric population include the sciatic nerve, with at least one case involving the musculature surrounding the nerve [23,24]. Researchers have also found that isolated symptoms such as chronic diarrhea and weight loss signify isolated relapse of ALL in the gut [25]. A complicating factor in gut relapse of ALL following a stem cell transplant is distinguishing GVHD from the relapse [26,27]. This is due to similar signs and symptoms of both GVHD and relapse, such as abdominal pain, diarrhea, and fever [26,27]. Similar to cardiac ALL relapse cases, biopsy of the involved tissue was required in all cases in those studies in order to make a definitive diagnosis via the identification of leukemic blasts [26,27]. Leukemic breast infiltration is also a rare phenomenon in the adult and pediatric ALL populations [28]. Table 3 summarizes the pertinent literature which describes leukemia infiltration involving various organs.

The microenvironments of sites of leukemia relapse are believed to attract leukemia cells and provide sustenance for their growth [32,33]. This also provides another aspect of cell metastasis to target via preventing the migration of leukemia cells to a niche with a favorable microenvironment [34]. This can be accomplished by inhibiting leukemia cell adhesion and growth in the specific niche similar to that accomplished by the use of methotrexate in cases of central nervous system leukemia or targeting of leukemia cell signaling such as that in the transforming growth factor β pathway to prevent attraction of leukemia cells [32,34]. Authors have also reported evidence that leukemia cells themselves communicate with cells in the BM and extramedullary sites to produce a favorable microenvironment through cell-to-cell contact and adhesion, further increasing resistance to chemotherapy [32]. Moreover, leukemia cells may reside in sites where penetration by systemic chemotherapy is minimal when compared with that in other areas, such as the BM [32]. Among pediatric leukemia patients who have been declared disease-free via peripheral blood and BM examination, autopsies have shown organs infiltrated with leukemia cells despite the use of systemic chemotherapy [21].

The incidence rate for EMR ALL following HSCT has been as high as 29% [29]. Moreover, whereas HSCT may exert a profound graft-versus-leukemia (GVL) effect in the BM, this effect is not as forceful in extramedullary sites considering that the EMR-free survival rates in patients who experienced GVHD and those who did not were 77% and 78%, respectively, whereas the BM relapse (BMR)-free survival rates in these patients were 81% and 59%, respectively [30,35]. Although initially only a single site of EMR may be present following HSCT, multiple sites are usually discovered over time, sometimes only during autopsy [20,21,31]. In some cases, the majority of extramedullary ALL relapses occurred within 2 years following HSCT [20]. In a study showing that ALL relapse in the breast was followed by spread to other EMR sites, the leukemia cells behaved like solid tumors, metastasizing to distant sites or stimulating hematopoietic cells in other organs to adopt a malignant phenotype [36]. As described previously, this solid tumor like behavior may be due to the microenvironment of those EM sites, including the breasts, which attracts leukemia cells, promotes their growth, and may contribute to the chemoresistance of leukemia cells at those sites [32,33,36]. Autopsy results have shown that pediatric leukemia patients who underwent HSCT but died while negative for relapse in the BM had a significant leukemia involvement in the kidneys, mediastinal soft tissue, adrenal glands, or intestines but with only moderate disease in the reticuloendothelial system [31]. Moreover, leukemia patients in that study who had relapses following HSCT entered remission after the emergence of GVHD. Researchers demonstrated that the role GVHD plays in inducing leukemia remission may be an indicator of a GVL effect [30]. BM remission in the setting of GVHD and leukemia infiltrates at extramedullary sites indicated GVL killing in the BM but not other organs or sites [30,31]. Additionally, in the patients in those studies, engrafted donor lymphocytes were cytotoxic to the recipient leukemia cells, indicating that leukemia remission is robust in lymphocyte-rich areas such as the BM and reticuloendothelial system due to the cytotoxic effects of donated lymphocytes. However, this leaves lymphocyte-poor areas such as the kidneys and intestines vulnerable to leukemia relapse [31].

## 4. Immunotherapy for Relapsed and/or Refractory Pediatric ALL

Novel targeted therapeutics such as bispecific antibodies and CAR T cells have the potential to improve outcomes in cases of relapsed and/or refractory (r/r) pediatric leukemia. The use of these new therapies has produced better outcomes than has salvage chemotherapy, with remission rates for CAR T-cell therapy that are almost twice as high as for those of conventional chemotherapy [37].

Blinatumomab is a bispecific T-cell engager that targets CD19-expressing leukemia cells. When used in the postinduction chemotherapy setting, blinatumomab has improved event-free and overall survival rates in pediatric patients with first relapse of high-risk B-cell ALL [37,38,39,40]. Because of the targeted effect of this drug, its use has been associated with a low risk of adverse effects while increasing overall survival rates and leading to remission in patients bridged to HSCT [41].

Inotuzumab ozogamicin is an antibody directed against CD22 and conjugated with the cytotoxin calicheamicin whose use leads to uptake and killing of leukemia cells that highly express CD22. In trials, inotuzumab ozogamicin has proven to be effective in inducing remission of r/r ALL with minimal adverse effects in the majority of patients [37]. The benefits of limited toxicity and success in treating MRD owe to the drug’s specific targeting of the CD22 surface receptor of leukemia cells [42]. Because inotuzumab ozogamicin was successful in treating r/r B-cell ALL in the phase 2 Children’s Oncology Group AALL1621 trial, the drug is now in phase 3 trials to determine its efficacy in the treatment of newly diagnosed high-risk B-cell ALL [43].

## 5. Immunotherapy and Transplantation in the Setting of EMR

Whereas HSCT, CAR T-cell therapy, blinatumomab, and inotuzumab ozogamicin are commonly used for treatment of r/r medullary leukemia, common treatments of EMR of leukemia are lacking [44]. In cases of leukemia EMR in the pediatric population, including after transplantation, patients have undergone conventional chemotherapy and, in certain instances, surgery or radiation therapy [2,12,24,35,45]. Generally, a combination of local and systemic chemotherapy has been effective in producing long-term survival in cases of relapse involving various regions, such as the breast, serosa, and ovary [20]. Modalities such as irradiation and DLI in combination with systemic chemotherapy have been successful in inducing repeat remission [20].

HSCT has been successful in curing isolated EMR, in some instances resulting in 10-year overall survival rates nearing 60% and disease-free survival rates of about 50% [46,47]. Moreover, different sources of donor stem cells, including matched family and matched unrelated donors, have not altered outcomes of EMR, which may have rather been related to the effectiveness of the conditioning regimens [46,48]. This allows for stratification of patients with BMR or EMR and thus a high risk of subsequent relapse, who may be offered allogeneic transplantation. In comparison, those with isolated EMR and a lower risk of subsequent relapse can be offered autologous transplantation, resulting in shorter treatment durations and lower risk of transplant-related morbidity and mortality [46].

However, authors have reported that immunotherapies such as CAR T-cell therapy can be effective in eradicating extramedullary disease [28,49,50]. CAR T-cell therapy is an exciting treatment modality to target leukemia cells. A patient’s T lymphocytes can be genetically modified to harbor a cell surface receptor that recognizes CD19-expressing leukemia cells, leading to cancer cell-specific cytotoxicity [51].

The effectiveness of CAR T cells in treatment of EMR of leukemia may lie in their ability to travel to extramedullary sites as well as remain in the BM and peripheral blood [50]. CAR T cells can also traverse the blood–brain barrier, leading to eradication of central nervous system ALL, which may indicate a similar ability to travel to extramedullary sites where leukemia cells may find safe harbor [52,53]. Other sites where CAR T cells have been believed to travel to and induce leukemia remission include extramedullary sites such as the genitourinary and intraocular systems [54,55]. The effectiveness of CAR T-cell therapy may be positively influenced by a low disease burden at the time of cell infusion [56,57]. Isolated EMR has occurred following CAR T-cell therapy, however [58].

Overall, standard guidelines for the treatment of EMR of ALL are lacking, although irradiation and combination chemotherapy are associated with improved outcomes [47,59,60]. Following HSCT, GVHD is suggested to be protective against isolated EMR, whereas certain cytogenetic markers, such as t(9;22) and t(8;21), and certain preconditioning regimens are associated with increased risk of EMR [30,59,60,61]. In addition, patients who received cyclophosphamide and busulfan prior to transplantation had a higher risk of EMR than did those who received a regimen including total-body irradiation [30].

Whereas CAR T-cell therapy may be somewhat effective in eradicating EM ALL, blinatumomab has not demonstrated a similar ability to induce remission in EM ALL due to either resistance to the drug or the emergence of a CD19-negative leukemia cell line [56,62,63]. A high disease burden and prior extramedullary disease have been associated with poor outcomes of blinatumomab use [64]. In patients with extramedullary and BM leukemia, blinatumomab has been effective in eradicating BM disease but similar outcomes have not been seen in those with EM ALL [64]. Regardless, resistance to blinatumomab does not predict response to CAR T-cell therapy [56].

Although data regarding inotuzumab ozogamicin use for EM ALL are limited, preliminary data demonstrated that it may have a role in treatment and eradication of extramedullary disease [65,66,67]. DLI following HSCT and in conjunction with radiation therapy has been somewhat effective in inducing sustained remission following post-HSCT relapse [68,69,70]. DLI induces a form of GVL effect but carries some risk of GVHD; also, knowledge of the specific antigens on leukemia cells that lymphocytes interact with to induce a cytotoxic effect is minimal [71]. Considering that in some cases, ALL remission is induced via the GVL effect in the absence of GVHD, the effect may be on minor histocompatibility antigens [71]. In cases in which DLI did not induce remission at sites of EMR, failure to recruit activated donor T cells to all organs harboring leukemia cells, poor lymphocyte activation, and failure to recruit accessory cells with antileukemia effects were believed to be the major culprits [72].

## 6. Immunotherapy Targeting Molecular Pathways

There are multiple molecular pathways, including the interleukin-7 receptor α (IL-7Rα) signaling pathway, that have been identified as possible targets of immunotherapy [73]. Aberrant signaling due to gain of function mutations in the *IL-7R**α* gene has been shown to promote the survival of T-cell ALL (T-ALL), especially when associated with cytokine receptor-like factor 2 (*CRLF2*), another gene mutation associated with pediatric ALL [74,75]. Considering that mutations in the *IL-7R**α* gene have been found in nearly 10% of pediatric leukemia cases, this pathway represents a fruitful opportunity for the development of an immunotherapy modality to target [76]. IL-7 and IL-7R*α* have also been found to interact with Janus kinase (JAK), leading to JAK phosphorylation and activation of downstream pathways such as signal transducer and activation proteins (STAT5) that can further lead to the proliferation of leukemia cells [77]. Each component represents multiple points that are targetable aspects in leukemia treatment and drugs such as ruxolinib for JAK/STAT inhibition and everolimus for mammalian targeting of rapamycin (mTOR) inhibition are in clinical trials to determine their utility in pediatric ALL [78]. Another pathway that has been identified in the development of ALL includes the phosphatidylinositol 3-kinase (PI3K)/AKT/mTOR pathway, which is known to promote the survival and proliferation of leukemia cells, especially in T-ALL [77]. Similar to aberrant IL-7R*α* signaling, the PI3K/AKT/mTOR and JAK/STAT pathways are associated with rearrangements of the *CRLF2* gene, leading to the development of Philadelphia-like (Ph-like) ALL, which can portend a poorer prognosis compared to those without the mutation [78,79]. Drugs such as ruxolinib which can inhibit STAT5 and JAK may have a role in the treatment of patients with Ph-like ALL that is positive for *CRLF2* mutation [78]. Due to the interactions of multiple different pathways, the targeting of one pathway may lead to a domino effect which inhibits multiple downstream molecular targets and leads to the inhibition of ALL proliferation.

## 7. Cardiotoxicity of Immunotherapy

Despite the promising future of immunotherapy for EMR of ALL, cardiotoxicity is a risk with modalities such as blinatumomab and CAR T-cell therapy. Both are associated with CRS, which is the most common adverse event in the pediatric population receiving CAR T-cell therapy for r/r ALL, in some cases affecting 85% of patients and with nearly half of them needing intensive care [80,81]. Depending on the severity of CRS, it can lead to significant shock requiring treatment with multiple pressor agents, refractory hypotension, and multiorgan dysfunction syndrome due to inadequate perfusion [82]. For a patient with disease limiting the heart’s function, CRS may be a deadly adverse event due to the inability of the heart to compensate for shock, vascular leakage, and hypoperfusion to major organs. The cardiotoxic effects specifically associated with CAR T-cell therapy, such as sinus tachycardia, severe decline in LV ejection fraction/LV systolic dysfunction, hypotension leading to shock, and electrocardiogram abnormalities, may be difficult to tolerate in a patient with leukemia involving the heart and hints that CAR T-cell therapy is a potentially unsafe treatment modality for patients with poor cardiac function [3,83]. In patients with cardiac dysfunction while experiencing severe CRS due to CAR T-cell therapy, physicians have used steroids and tocilizumab to support organ function while preserving the effectiveness of the CAR T cells [83].

Patients with ALL involving the heart also have a high risk of morbidity and mortality from immunotherapy due to prior exposure to cardiotoxic chemotherapy and radiation [84]. Medications such as anthracyclines and cyclophosphamide, which are mainstays of pediatric lymphoblastic leukemia treatment, can lead to congestive heart failure, LV dysfunction, and arrhythmia during and after treatment [85]. In patients with radiation exposure, vascular disease and arrhythmias may also occur [84]. Furthermore, patients with a history of heavy pretreatment are believed to lack the cardiac reserve needed to compensate for the sequalae of CRS [83].

Taking into account the cumulative effects of prior chemoradiation and the risk of cardiotoxicity from immunotherapeutic modalities such as blinatumomab and CAR T-cell therapy, patients who undergo these treatments are at significantly high risk for adverse effects due to cardiac events. The use of immunotherapy in these patients requires deliberation of over the risks and benefits of immunotherapy and appropriate diagnostic steps, such as an echocardiogram to measure LV ejection fraction, LV global longitudinal strain, and levels of cardiac biomarkers to evaluate and monitor the heart’s ability to compensate for negative effects that may arise due to adverse effects of immunotherapy, especially during episodes of CRS [83].

## 8. Prognosis Following EMR of ALL

Although isolated EMR following stem cell transplantation portends a poor prognosis for ALL, it is not significantly worse than that with BMR, with studies demonstrating that it is associated with better survival and a greater chance of cure [59,86,87]. In one study, the median survival times following relapse were 11 months for EMR but only 2 months for BMR [29]. This is due in part to advances in the early diagnosis and combination treatment of r/r ALL, including in those who experience relapse following allogeneic HSCT [28,45,56,88,89,90]. Factors associated with increased risk of EMR include chronic GVHD and a longer duration from stem cell transplantation relapse than that associated with BMR [29]. The elevated risk of EMR in the setting of chronic GVHD provides further evidence of a stronger GVL effect in the BM than in extramedullary sites. However, authors reported that patients with acute GVHD following transplantation for EMR of ALL had higher relapse-free survival than did those who did not experience acute GVHD [30]. The incidence of EMR late after transplantation may also indicate that the GVL effect is more effective in preventing early than late relapse [29]. Moreover, in another study, patients with ALL who relapsed later following stem cell transplantation were more likely to experience relapse in the form of isolated EMR followed by patients with both EMR and BMR, and lastly, those with BMR only [30]. Patients who initially have EMR followed by BMR, including those with prior BMR, have worse prognoses than do those with isolated EMR [91]. Despite negative MRD, leukemia cells at sites of EMR may also exhibit homology with the original leukemia clone [92]. This demonstrates that initial chemotherapy may have only decreased leukemia burden at the BM to an undetectable level and that cells in “sanctuary sites” may have escaped the cytotoxic effects of chemotherapy [92].

## 9. Conclusions

We present herein the first documented biopsy-proven case of post-HSCT symptomatic B-cell ALL relapse within the myocardium that does not also involve the BM in a pediatric patient. Although patients with isolated EMR fare better than do those with EMR in combination with medullary relapse or isolated medullary relapse, our patient was stable for 8 months following HSCT before dying of cardiac failure. Isolated symptomatic cardiac relapse of ALL may be difficult to diagnose due to the location of disease, the need for invasive procedures required for accurate diagnosis, and care requiring cardiac critical care services. Currently, uniform guidelines for treating isolated EMR of ALL are lacking, although combination chemotherapy regimens with or without radiation therapy and HSCT have been somewhat effective. Although CAR T-cell therapy has been effective to a certain extent in controlling EMR, its use for isolated cardiac relapse is questionable due to a significant risk of adverse events related to its cardiotoxicity and inability of the heart to compensate for the sequelae of CRS. While our patient was unable to receive immunomodulating therapies such as blinatumomab or CAR T-cell therapy because these treatments had not yet been approved for pediatric use during the time of his illness, immunotherapies such as CAR T-cell therapy, T-cell-engaging antibodies, and monoclonal antibodies appear to be effective in treating EMR of ALL. These immunotherapies are potential modalities of cancer treatment, and studies suggest that they are treatment options that can target leukemia cells in areas of the body where conventional chemotherapy and HSCT have little to no effect on them.

## Figures and Tables

**Figure 1 cancers-13-05814-f001:**
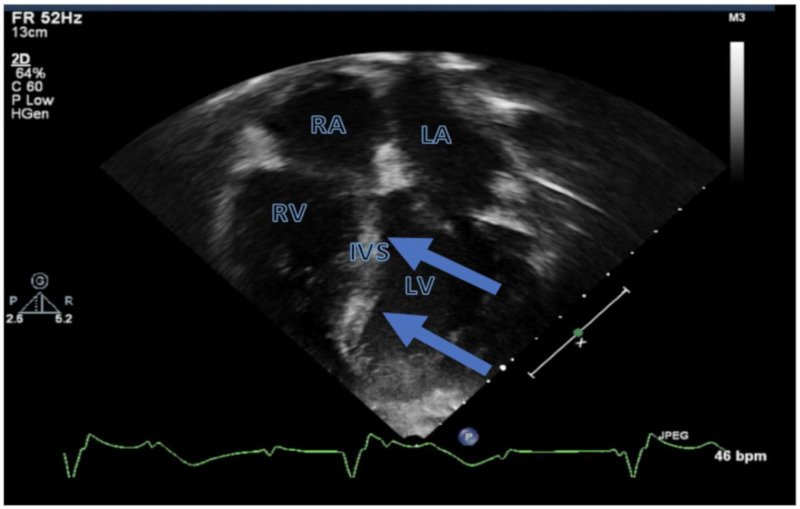
Echocardiogram of the patient on the day of presentation. The arrows point to echo-bright areas over the interventricular septum indicating leukemia cell infiltration. RA, right atrium; RV, right ventricle; LA, left atrium; LV, left ventricle; IVS, intraventricular septum.

**Figure 2 cancers-13-05814-f002:**
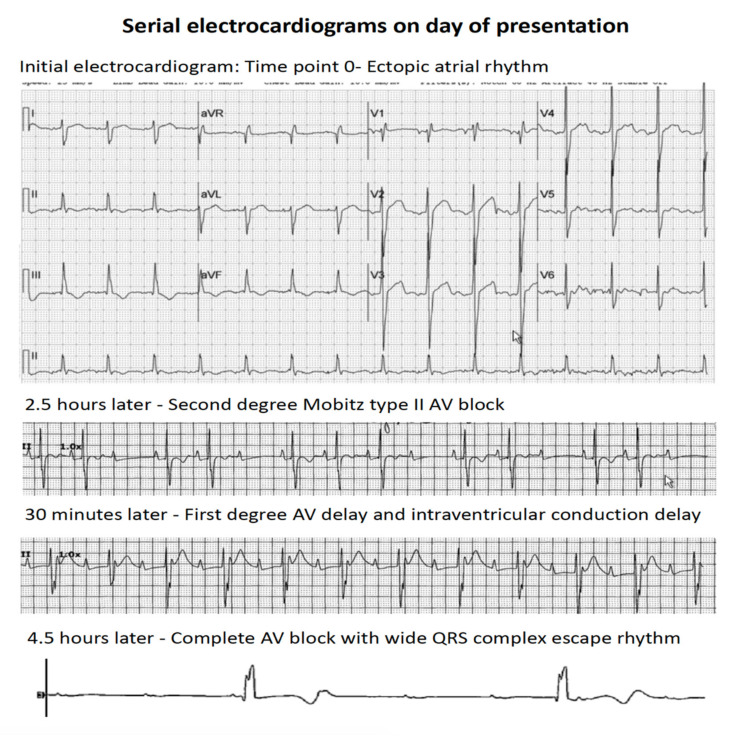
Serial electrocardiograms of the patient on the day of presentation demonstrating progressive AV conduction blockage.

**Figure 3 cancers-13-05814-f003:**
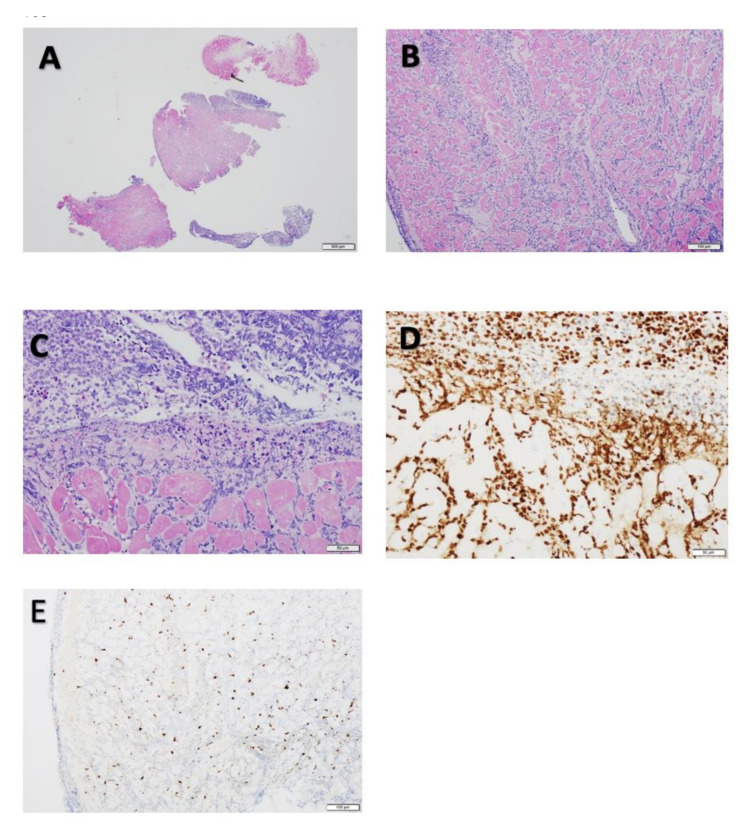
(**A**) Hematoxylin and eosin stain (low-power view, scale: 500 µM ) showing dense lymphocytic infiltrates involving every piece of the tissue biopsy sample and endocardium. (**B**) Hematoxylin and eosin stain of the dense lymphocytic infiltrates (midpower view, scale: 100 µM). The infiltration is widespread around each myocyte. (**C**) Hematoxylin and eosin stain of the infiltrating cells (high-power view, scale: 50 µM) showing highly malignant features, including numerous apoptotic bodies, frequent mitoses, and myocyte necrosis. (**D**) Immunohistochemical stain for PAX-5 demonstrating diffuse and strong positivity for the marker (higher-power view, scale: 50 µM). (**E**) Immunohistochemical stain for TdT, a marker for diagnosis of acute leukemia. About 10% of the tumor cells were positive for TdT, supporting the diagnosis ALL. No features of sarcoidosis or amyloidosis are shown (midpower view, scale: 100 µM).

**Figure 4 cancers-13-05814-f004:**
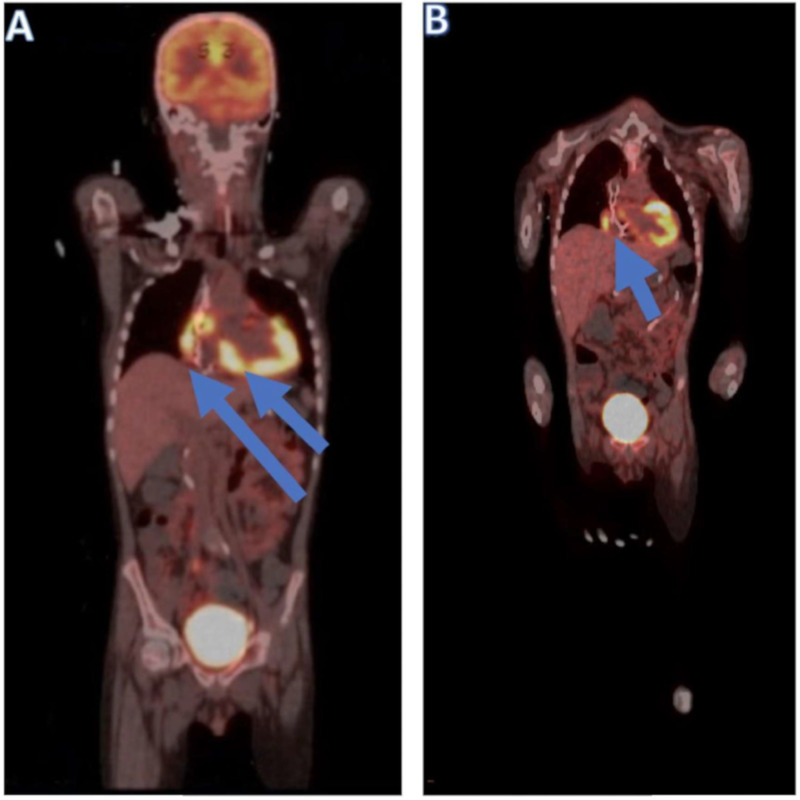
PET scans of the patient showing ^18^F-fluorodeoxyglucose uptake in the myocardium involving the right and left ventricles. (**A**) The arrows indicate abnormal uptake in the myocardium involving both ventricles, with a maximum standard uptake value of 15. This uptake is not a typical finding as leukemia in consideration of the high metabolic activity of the lesions. (**B**) The arrows indicate persistent hypermetabolism involving the myocardium predominantly in the lateral wall of the right atrium, the right ventricle, and into the ventricle septum, with a maximum standard uptake value of 8.7.

**Figure 5 cancers-13-05814-f005:**
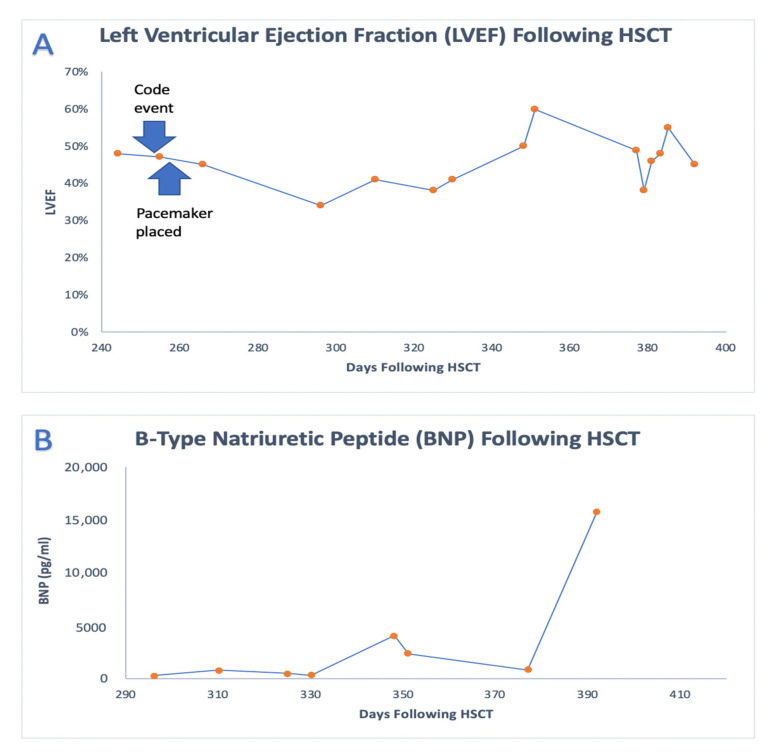
The trends of the left ventricular ejection fraction (LVEF) and B-type natriuretic peptide (BNP) following HSCT are detailed. (**A**) The post-HSCT trend in LV ejection fraction (LVEF) following the patient’s code event and eventual pacemaker placement. (**B**) The patient’s post-HSCT B-type natriuretic peptide (BNP) level, which increased during times of acute cardiac failure.

**Table 1 cancers-13-05814-t001:** Summary of the patient’s clinical characteristics and lab findings upon diagnosis. Abbreviations: CT, computed tomography; FISH, fluorescence in situ hybridization; MLL, mixed lineage leukemia; TdT, terminal deoxynucleotidyl transferase; WBC, white blood cell.

Physical Exam	Peripheral Blood Labs	Bone Marrow Aspiration and Biopsy	Genetic Markers	Cerebrospinal Fluid Analysis	Imaging
Initial concerning signs: right thigh and cheek pain and numbness Difficulty opening mouth Mild hepatomegaly Decreased sensation on the face in cranial nerve V2–V3 distribution, right facial droop, extraocular movements intact No testicular masses palpated	White blood cells: 70,000 cells/μL Hemoglobin: 7.2 g/dL Platelets:11,000 cells/μL Peripheral blasts: 83%	94% blasts, CD10^+^, CD19^+^, CD20^+^ [variable positive], CD22^+^, CD34^−^, CD38^+^ [decreased but positive], CD45^−^ Hyper-diploid blasts identified No T-cell surface markers identified Negative for TdT and myeloperoxidase	Blasts with chromosome 6q deletion Negative for FISH BCR-ABL fusion protein, *MYC* gene rearrangement, *MLL* gene rearrangement	Blasts in the presence of peripheral blood detected	CT scan of the head: no intracranial abnormalities with preservation of gray-white differentiation

**Table 2 cancers-13-05814-t002:** Summary of the patient’s treatment regimens and associated complications.

Time Period Following Initial B-ALL Diagnosis	Event	Treatment	Complications
Initial diagnosis		COG AAL1131 very high-risk arm with CNS involvement IT cytarabine IT methotrexate	Persistent numbness of right cheek CN deficits: CN VI and VII palsy, CN V dysfunction (tooth pain); lumbar radicular pain
3 months	Symptomatic worsening of neurological disease	Oral dexamethasone IV vincristine IV PEG asparaginase	Marrow relapse (15% blasts in peripheral blood and 81% leukemic blasts in BM)
5 months	Very early combined relapse	Reinduction chemotherapy with hyper-CVAD IV cyclophosphamide IV vincristine IV doxorubicin Oral dexamethasone IT cytarabine High-dose IV methotrexate High-dose IV cytarabine IT methotrexate IV rituximab	Persistent left-sided motor weakness of the mouth
7 months	HSCT	Preparative regimen IV fludarabine IV cyclophosphamide TBI at a total dose of 1200 cGy (150-cGy fractions twice daily on days −7 to −4) Stem cell transplant dose: 3.8 × 10^7^ total nucleated cells/kg	Grade II acute GVHD of the upper and lower GI tract
16 months	Post-transplantation relapse 1	R-MOAD IV rituximab IV methotrexate IV vincristine IV PEG asparaginase Oral dexamethasone	Symptomatic cardiac relapse
22 months	Refractory disease	IT cytarabine IV etoposide IV cyclophosphamide	Death due to severe LV systolic dysfunction and loss of AV conduction

Abbreviations: ALL, acute lymphoblastic leukemia; AV, atrioventricular; BM, bone marrow; cGY, cobalt gray; COG, Children’s Oncology Group; CNS, central nervous system; IT, intrathecal; CN, cranial nerve; IV, intravenous; GI, gastrointestinal; GVHD, graft versus host disease; HSCT, hematopoietic stem cell transplant; LV, left ventricle; MLL, mixed-lineage leukemia; TBI, total-body irradiation

**Table 3 cancers-13-05814-t003:** Literature describing leukemic infiltration of various organs.

Author	Disease	Summary of Findings
Roberts et al. (1968) [9]	Leukemia involving the heart	420 patients in the study of postmortem examination of hearts of deceased patients with leukemia. 60% patients in the study had ALL and 40% had AML; 37% of hearts found to have leukemic infiltrates; infiltrates described as focal and few in number; mainly in LV and RA.
Sumners et al. (1969 ) [10]	Leukemia involving the heart	116 hearts of children studied over a 5-year period. 44% patients found to have leukemic infiltrates of the heart
Hori et al. (2006); Malbora et al. (2010); and Barbaric et al. (2002) [2,11,12]	Systemic leukemia involving the heart	Adolescents presenting with cardiac symptoms related to leukemic involvement of the heart; echocardiogram and chest imaging may help with diagnosing leukemic involvement of the heart
Wiernik et al. (1976); Hardikar et al. (2002); Espino et al. (2001) [16,17,18]	Systemic leukemia involving the heart	Signs and symptoms of cardiac involvement of leukemia: respiratory symptoms, pleural or pericardial effusions, and cardiomyopathy; require distinguishing from treatment related adverse effects
Nies et al. (1965) [21]	Leukemia with various organ involvement	Infiltration of various organs including kidney (most common), followed by liver, testes, and bowel
Cunningham et al. (2006); Ragoonanan et al. (2021) [22,28]	Systemic leukemia involving the breast	Sites of infiltration can include the breast; can represent a site of resistant leukemia including ALL and AML; leukemic infiltration of the breast following HSCT and CAR T-cell therapy has also been described
Bhatti et al. (2010); Kletzel et al. (2000); Papadakis et al. (2010); [25,26,27]	Systemic leukemia involving the GI tract	Relapsed leukemia in pediatric patients involving the GI tract; presentation can mimic acute GVHD when occurring following HSCT; symptoms can be nonspecific including abdominal pain and diarrhea
Chong et al. (2000); Au et al. (1999); Lee et al. (2005); Odom et al. (1978) [29,30,31]	EMR following HSCT	Rate of relapse at extramedullary sites following HSCT: 29%; areas of relapse following HSCT kidneys, mediastinal soft tissue, adrenal glands, intestine; GVL effect may be more forceful in the bone marrow rather than EM sites and those that are lymphocyte poor

Abbreviations: ALL, acute lymphoblastic leukemia; AML, acute myeloid leukemia; EMR, extramedullary relapse; GI, gastrointestinal; GVL, graft-versus-leukemia; HSCT, hematopoietic stem cell transplant; LV, left ventricle; RA, right atrium.

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
