# Peer review of "Cardiac Relapse of Acute Lymphoblastic Leukemia Following Hematopoietic Stem Cell Transplantation: A Case Report and Review of Literature"

_cancers, 2021, doi:10.3390/cancers13225814_

Round 1
Reviewer 1 Report
The Manuscript by Irtiza N. Sheikh et al. describes a B-ALL case study and reports the associated literature.
The Manuscript could be improved: authors could add a table reporting the clinical characteristics of the patient at diagnosis, including (if available) data on gene mutations. As for Table 1, authors could add a column reporting the timing of the reported event.
As for the other paragraphs, authors could add a table resuming the most relevant data reported in paragraph 3 (lines 208-243; lines 259-282 for HSCT) and the related studies. They could also add a percentage for "the vast majority of these patients" (line 211) and "the child's age" (line 218).
Moreover, authors could also discuss about recent data on new molecular targets in ALL that could be relevant for new treatments, like the ones presented in the following papers:
1: Rodrigues GOL, Cramer SD, Winer HY, Hixon JA, Li W, Yunes JA, Durum K. Mutations that collaborate with IL-7Ra signaling pathways to drive ALL. Adv Biol Regul. 2021 May;80:100788. doi: 10.1016/j.jbior.2021.100788. Epub 2021 Jan 21. PMID: 33578108.
2: Paganelli F, Lonetti A, Anselmi L, Martelli AM, Evangelisti C, Chiarini F. New advances in targeting aberrant signaling pathways in T-cell acute lymphoblastic leukemia. Adv Biol Regul. 2019 Dec;74:100649. doi:10.1016/j.jbior.2019.100649. Epub 2019 Sep 5. PMID: 31523031.
In addition, they could also discuss about other recent reports questioning the relevance of targeted therapy in pediatric B-ALL (Ratti S, Lonetti A, Follo MY, Paganelli F, Martelli AM, Chiarini F, Evangelisti C. B-ALL Complexity: Is Targeted Therapy Still A Valuable Approach for Pediatric Patients? Cancers (Basel). 2020 Nov 24;12(12):3498. doi:10.3390/cancers12123498. PMID: 33255367; PMCID: PMC7760974)
Finally, authors should carefully revise the English language and format of the paper, as there are a few of them. For instance:
line 232: "[19].Areas" should be "[19]. Areas"
line 241: "in" is bold but it should be normal
line 327: between "cells," and "leading" there is an extra-space
line 397: "levels of" is bold but it should be normal
Author Response
We appreciate the opportunity to revise our manuscript based on the thoughtful comments provided by the reviewers. Our point-by-point changes to the manuscript are detailed below based on each reviewers’ comments:
Reviewer 1:
- The Manuscript could be improved: authors could add a table reporting the clinical characteristics of the patient at diagnosis, including (if available) data on gene mutations. As for Table 1, authors could add a column reporting the timing of the reported event.
--Thank you. From lines 96-98: A table has been added which summarizes the patient’s clinical characteristics as well as lab findings when the patient was diagnosed. The gene mutations that were available have also been added.
--Thank you. At line 100, table 1 has been re-labeled as table 2 and as the reviewer suggests, time periods for each event have been added as related to the time after the initial diagnosis.
- b) As for the other paragraphs, authors could add a table resuming the most relevant data reported in paragraph 3 (lines 208-243; lines 259-282 for HSCT) and the related studies. They could also add a percentage for "the vast majority of these patients" (line 211) and "the child's age" (line 218).
--Thank you. A new table has been added (Table 3. Literature describing leukemic infiltration of various organs) starting at line 258 in order to summarize the pertinent findings of the various studies that describe leukemia infiltration of different organs as well as studies that describe infiltrative relapse following HSCT.
On line 222, the percentage has been added instead of “the vast majority of these patient.”
--on line 230, the age and gender of the patient have been added as well
- c) Moreover, authors could also discuss about recent data on new molecular targets in ALL that could be relevant for new treatments, like the ones presented in the following papers:
1: Rodrigues GOL, Cramer SD, Winer HY, Hixon JA, Li W, Yunes JA, Durum K. Mutations that collaborate with IL-7Ra signaling pathways to drive ALL. Adv Biol Regul. 2021 May;80:100788. doi: 10.1016/j.jbior.2021.100788. Epub 2021 Jan 21. PMID: 33578108.
2: Paganelli F, Lonetti A, Anselmi L, Martelli AM, Evangelisti C, Chiarini F. New advances in targeting aberrant signaling pathways in T-cell acute lymphoblastic leukemia. Adv Biol Regul. 2019 Dec;74:100649. doi:10.1016/j.jbior.2019.100649. Epub 2019 Sep 5. PMID: 31523031.
In addition, they could also discuss about other recent reports questioning the relevance of targeted therapy in pediatric B-ALL (Ratti S, Lonetti A, Follo MY, Paganelli F, Martelli AM, Chiarini F, Evangelisti C. B-ALL Complexity: Is Targeted Therapy Still A Valuable Approach for Pediatric Patients? Cancers (Basel). 2020 Nov 24;12(12):3498. doi:10.3390/cancers12123498. PMID: 33255367; PMCID: PMC7760974)
--Thank you. We agree that these studies as well as the discussion on molecular targets would improve our manuscript. For that reason, starting at line 390, we have dedicated a section to “Immunotherapy targeting molecular pathways.” In that section we describe the studies highlighted by the reviewer as well as add other studies which describe the molecular pathways in ALL that may be targetable by immunotherapy. Moreover, the study by Ratti et al., described molecular pathways targeted by specific drugs and we have included that discussion as indicated by citation number 77 on lines 404, 410, and 412.
- d) Finally, authors should carefully revise the English language and format of the paper, as there are a few of them. For instance:
line 232: "[19].Areas" should be "[19]. Areas".
line 241: "in" is bold but it should be normal.
line 327: between "cells," and "leading" there is an extra-space
line 397: "levels of" is bold but it should be normal
--Thank you. The paper has been revised for language and format extensively. We have made the above revisions and others throughout the paper to ensure that English is used correctly and is readable.
Reviewer 2 Report
Dr. Irtiza N. Sheikh DO et al. described the case in detail and provided related reviews. The interpreting is fairly good, but the two parts are not combined closely. And the reviewer wants the authors can provide more details on the tissues’ examination of PB/BM before treatment, like staining.
Author Response
We appreciate the opportunity to revise our manuscript based on the thoughtful comments provided by the reviewers. Our point-by-point changes to the manuscript are detailed below based on each reviewers’ comments:
Reviewer 2: Dr. Irtiza N. Sheikh DO et al. described the case in detail and provided related reviews. The interpreting is fairly good, but the two parts are not combined closely. And the reviewer wants the authors can provide more details on the tissues’ examination of PB/BM before treatment, like staining.
--Thank you for your suggestion. We agree that additional details on tissue examination including staining before treatment would add great value to this study. However, because the patient presented more than 5 years prior to writing this manuscript, the pathologist was unable to retrieve an extensive amount of prior stains from the tissue repository. For that reason, we attempt to describe as much of the patient’s clinical and lab findings at the time of diagnosis, including adding a new table (table 1) which describes molecular and genetic findings on bone marrow as suggested by reviewer 1.
--In terms of the two parts (i.e case presentation and related reviews), our purpose was to describe the rarity of isolated cardiac relapse as well as the modern methods currently for possible treatment. This is in contrast to the treatment the patient received, because at the time of diagnosis, immunotherapies such as blinatumomab and CAR T-cell therapy were not approved by regulatory agencies (FDA) for the pediatric population.
Reviewer 3 Report
Sheikh I.N. et al. presented a very rare pediatric case with cardiac relapse of ALL, including a review of the literature.
Major Comment
It is not adequately explained the reason why the authors did not use immunotherapy (Blinatumomab or Inotuzumab) to treat this type of relapse, occurred after hematopoietic stem cell transplantation (HSCT). The authors widely explored the different presentations and origins of ExtraMedullary Relapses (EMRs), but it is clear that this case showed a chemotherapy resistant disease. Thus, after HSCT, the adoption of immunotherapy would have been inevitable, although the high risk of immunological side effects involving indirectly the cardiac functions.
Moreover Authors pointed out several issues. An interesting one is the "solid tumor like behavior" of ALL. This concept is strictly related to microenvironment where lymphoblastic leukemia cell grows and metastasizes. Please clarify this crucial issue.
Author Response
We appreciate the opportunity to revise our manuscript based on the thoughtful comments provided by the reviewers. Our point-by-point changes to the manuscript are detailed below based on each reviewers’ comments:
Reviewer 3
Major Comment
- It is not adequately explained the reason why the authors did not use immunotherapy (Blinatumomab or Inotuzumab) to treat this type of relapse, occurred after hematopoietic stem cell transplantation (HSCT). The authors widely explored the different presentations and origins of Extra Medullary Relapses (EMRs), but it is clear that this case showed a chemotherapy resistant disease. Thus, after HSCT, the adoption of immunotherapy would have been inevitable, although the high risk of immunological side effects involving indirectly the cardiac functions.
--Thank you. We agree that the patient demonstrated chemotherapy resistant disease and may have benefitted from immunotherapy. However, at the time that the patient presented (> 5 years prior to writing this manuscript), immunotherapies such as CAR T-cell, blinatumomab were not available as they had not been yet approved for pediatrics (and there was very limited data for the use of inotuzumab for pediatric patients). In line 487, we describe that the patient was not able to receive immunotherapy due to the lack of pediatric indication or limited data available at that time.
- b) Moreover Authors pointed out several issues. An interesting one is the "solid tumor like behavior" of ALL. This concept is strictly related to microenvironment where lymphoblastic leukemia cell grows and metastasizes. Please clarify this crucial issue.
--Thank you. We truly appreciate the opportunity for this clarification. For that reason, on line 290, we have explained that the microenvironment of the extramedullary sites (including the breast) are responsible for the solid tumor like behavior in terms of metastasis. We have also included references to support those observations.
Round 2
Reviewer 2 Report
The authors addressed my concerns.